

# Performance of snow density measurement systems in snow stratigraphies

Jiansheng Hao[1,2,3,4], Farong Huang[1,2,3,4,5], Ditao Chen[1,2,3], Shuyong Mu[6], Lanhai Li[1,2,3,4,5]

[1]State Key Laboratory of Desert and Oasis Ecology, Xinjiang Institute of Ecology and Geography, Chinese Academy of Sciences, Urumqi 830011, China

[2]Ili Station for Watershed Ecosystem Research, Chinese Academy of Sciences, Xinyuan 835800, China

[3]University of Chinese Academy of Sciences, Beijing 100049, China

[4]CAS Research Center for Ecology and Environment of Central Asia, Urumqi 830011, Xinjiang, China

[5]Xinjiang Key Laboratory of Water Cycle and Utilization in Arid Zone, Urumqi 830011, Xinjiang, China

[6]Xinjiang Regional Center of Resources and Environmental Science Instrument, Chinese Academy of Sciences, Urumqi 830011, China

*Correspondence to*: Lanhai Li (lilh@ms.xjb.ac.cn)

**Abstract**. Gravimetric and dielectric permittivity measuring systems are applied to measure snow

density, but few studies have addressed differences between the two measurement systems under complex snowpack conditions. A field experiment was conducted to measure snow density using the two measurement systems in different stratigraphical layers consisting of fragmented precipitation particles (DF), faceted crystals particles (FC), depth hoar (DH) and melt forms (MF), and the performance of measurement systems was analyzed and compared. The results showed that the

measured density from the gravimetric measurement system was significantly higher than that from the dielectric permittivity measurement system. The precision and accuracy of the gravimetric measurement system were higher than that of the dielectric permittivity measurement system in the DF, FC and DH layers, but the precision and accuracy of two measurement systems were similar in the MF layers. By comparing the precision and accuracy as well as merits and drawbacks of the two

measurement systems, it was concluded that using gravimetric measurement system during dry snow period and dielectric permittivity measurement system during wet snow period will help surveyors obtain more reliable data. Furthermore, the study provided an approach which will facilitate the integration of the data obtained from different studies with different measurement systems into global databases.

**1 Introduction**

     Snow cover is a critical component linking the global climate system and the earth surface system, and provides water resources to large populations worldwide (Huning and AghaKouchak, 2018; Skiles et al., 2018; Barnett et al., 2005; Sturm et al., 2002). Density is one of the fundamental and important



snow properties, which varies over time (Carrol, 1977; Conger and McClung, 2009; De Michele et al.,

2013). It plays a key role in shaping a wide range of snow properties and physical processes. Snow

mechanical parameters are determined and estimated based on its density (Jamieson and Johnston,

2001; Abe, et al., 2006; Wang and Baker, 2013; Hannula et al., 2016). Permeability, photochemistry

and thermal conductivity are linked to density and depend on vertical density variations (Sturm et al.,

1997; Calonne et al., 2011, 2012). Snow density is also an indispensable input parameter for snow

dynamic models such as SNOWPACK (Lehning et al., 2002) and CROCUS (Brun et al., 1989). Snow

density has many applications in hydrological cycle studies (Sturm et al., 2010), water resources

management (Barnett et al., 2005; Huning and AghaKouchak, 2018; Jonas et al., 2009), ecosystem

studies (Rixen et al., 2008), climatology studies (Okuyama et al., 2003) and avalanche forecasts

(Schweizer and Jamieson, 2001). It is therefore essential to determine snow density for cryospherical

and hydrological studies. A precise and standardized measurement of snow density is of major

importance to better understand snow dynamics. However, separate studies with similar aims often

used different measurement systems to obtain snow density, and snow density data from different

measurement systems in the same snowpack was significantly different (Hawley et al., 2008; Conger

and McClung, 2009; Proksch et al., 2016). Therefore, difficulties exist for using snow density data

from separate studies obtained by different measurement systems in international research and to be

integrated into global databases. Quality evaluation and assimilation of snow density data obtained by

different measurement systems are a necessary step before the integration of the data obtained from

separate studies with different measurement systems into global databases, but few studies have

addressed assimilation of snow density data from different measurement systems.

Snow density has been measured in the field and laboratories using various measurement systems.

These measurement systems include gravimetric measurement system, dielectric permittivity

measurement system, the neutron-scattering probe, micro-computed tomography and diffuse

near-infrared transmission (Hawley et al., 2008; Conger and McClung, 2009; Proksch et al., 2016;

Gergely et al., 2010). In the field, snow density data are normally measured by technical staff, and

measurements are made at flat study plots (Schweizer and Jamieson, 2001; Conger and McClung, 2009,

Proksch et al., 2016). Taking the cost of measuring equipment, the technical simplicity and the

efficiency of observation into consideration, gravimetric measurement system and dielectric

permittivity measurement system are often applied to the measurement of snow density in the field



(Hawley et al.et al. 2008; Conger and McClung, 2009; Wilhelms, 2005; Kinar and Pomeroy, 2015).

Gravimetric measurement system consists of two parts: high precision electronic balance and different style cutters with a given volume. To determine snow density, a core sample of snow is extracted from the snow profile using a snow sampler. The density of the sampled snow is determined by weighted snow mass divided by the cutter's volume with a unit of kg m$^{-3}$. Dielectric permittivity measurement system can measure the real and imaginary permittivity and attenuation of snow to determine snow

density. The "Finnish Snow Fork" has been widely used for these measurement (Sihvola and Tiuri, 1986; Sugiyama et al.et al., 2010; Harper and Bradford, 2003; Techel and Pielmeier, 2011). Carroll (1977) reported that there was no significant difference between 200cm$^3$ and 100cm$^3$ box-type cutters and 500cm$^3$ tube-type cutters for snow density measurement, and that inexperienced operators tended to overestimate the densities found by more experienced workers, which was attributed to snow grain

type and associated measurement difficulties. Hawley et al. (2008) evaluated the accuracy of gravimetric measurement system, dielectric permittivity measurement system and the neutron-scattering probe. Dielectric permittivity measurement system underestimated snow densities in lower-density snow but agreed with the gravimetric measurement system for higher-density snow (Hawley et al., 2008). Conger and McClung (2009) compared box-, wedge-, and cylinder-type density

cutters and reported that there was a variation of 3–12% between the three cutter types. Recently, Proksch et al. (2016) compared the density cutters to micro-computed tomography and found snow densities measured by different measurement systems agreed within 9%. Leppänen et al. (2016) reported that the snow density from the Snow Fork was lower than that from the gravimetric measurement system. These previous studies focused on evaluating and comparing the accuracy of

different measurement systems in same snowpack condition and analyzing the influence of measuring instrument errors and personnel errors on measurement accuracy. The existing literature also reported that snow properties have a significant influence on the accuracy and precision of density measurement systems. However, relatively few studies have conducted to explicitly evaluate and compare the accuracy of different systems under different snowpack conditions.

Snow crystals undergo low, high and isothermal temperature gradient metamorphism growth over time (Fierz et al. 2009). Having undergone different meteorological conditions, the structure in each layer evolves differently from adjoining layers in terms of grain shape, grain size, inter-granular bonding, density, hardness, wetness and so on. Therefore, snowpack is made up of many snow layers


with different physical properties, and snow stratigraphy is constantly changing during a snow cover

period. This is especially true in thick snowpack, where stratification boundaries of the snowpack are

obvious, and the characteristics of adjacent snow layers are significantly different (Harper and Bradford,

2003). Snow properties have a significant influence on the accuracy and precision of density

measurement system. Therefore, density measurement system shows different accuracy and precision

with the change of snow stratigraphy with different physical properties. Selection of measurement

systems based on the performance of different measurement systems in various snow stratigraphies will

help to obtain more reliable data and optimize field measurements. However, the performance of

different measurement systems in different snow stratigraphies is poorly understood. Although the

existing literature provides an overview of the merits and drawbacks of different measurement systems,

tangible guidance on how to make decisions based on measurement system selection in various snow

stratigraphies for users is also not clearly provided.

To better understand the performance of different measurement systems in various snow

stratigraphies, snow-pit measurements were carried out at the Tianshan Station for Snowcover and

Avalanche Research (TSSAR) in the winter of 2017/2018. Gravimetric measurement system and

dielectric permittivity measurement system were applied to measure snow density in various snow

stratigraphies. The objectives of this study are to assess (1) whether the same measurement system

showed similar performance in different snow stratigraphies, and whether the two measurement

systems provided similar results in the same snow stratigraphy; and(2)whether assimilation of the

density data obtained by different measurement systems is feasible. Precision and accuracy as well as

merits and drawbacks of the two measurement systems in terms of applicability (time and labour

needed) were discussed, and recommendations in terms of practicality for field measurement were

derived. The result will help field operators to choose more effective and reasonable measurement

system based on snowpack characteristics in the field. Meanwhile, assimilation of density data obtained

by different measurement systems will allow the data from different measurement systems to be more

widely used in international research and contribute to global snowpack databases.

**2 Data and Methods**

**2.1 Measurement systems of snow density**

The following section gives an overview of the instruments and methods which were used to

measure snow density in TSSAR during the winter of 2017/2018.





Gravimetric methods measurement system has the longest history and has proven utility. It

consists of two parts: high precision electronic balance and different styles cutters. The 100 cm$^3$

box-cutter with 6cm×5.5cm×3cm was used in this study (Fig. 1a), which is based on a design

originated by the Institute of Low Temperature Science, Japan. The box sampler is a rectangular frame

open at both ends. It has a handle on one end. The digital electronic weighing scale (from

http://snowmetrics.com/shop/prosnow-kit-i/) is plastic, portable and waterproof and measures up to

1000 g at a 0.1g resolution with accuracy of ±0.01g, under an operating environment of –25℃to

+40℃. The researchers used a snow shovel to dig a snow pit from the snow surface to ground level and

used a snow saw to obtain a profile of snowpack in the observation field. The weighing scale was then

placed on a flat surface and zeroed. The cutter pushed horizontally into the target layer being measured

to collect the sample, and all snow was cleaned from the outside of the sampler. The snow sample was

extracted and put in a plastic bag, and then put the sample on the weighing scale and recorded weight

data.

The Snow Fork is designed for measuring the properties of snow in the field (Fig. 1b) (Sihvola and

Tiuri, 1986; Toikka, 2009). It is light, quick, and simple to operate. It has already been used in many

studies and have shown good results (Tiuriand Sihvola, 1986; Harper and Bradford, 2003; Sugiyama et

al., 2010; Techel and Pielmeier, 2011). The Snow Fork is designed to operate in extreme conditions

with temperatures low to -25℃(Toikka, 2009). The Snow Fork is a portable instrument, consisting of

an electronics box, a sensor, a keyboard and rechargeable batteries. It probes samples of snow within a

cylindrical volume of about 2 × 7.5 cm and operates between 500 and 1000 MHz. The sensor of the

Snow Fork is a steel fork used as a microwave resonator. Resonant frequency, attenuation, and 3-dB

bandwidth are measured by the Snow Fork, and the results are used to accurately calculate snow

density and liquid water content. The details of instrument operation can be found at

https://toikkaoy.com/.

**2.2 Stratigraphy and mechanical properties of snow**

Snowpack is made up of many snow layers with different physical properties, and the stratification

boundaries of snowpack are obvious. A stratigraphic layer is a certain stratum with similar properties

(size of the grains, microstructure,shear strength, hardness) in the snowpack according to Fierz et al.

(2009). An overview of the methods which were used to measure these snow properties is described in

the following section.



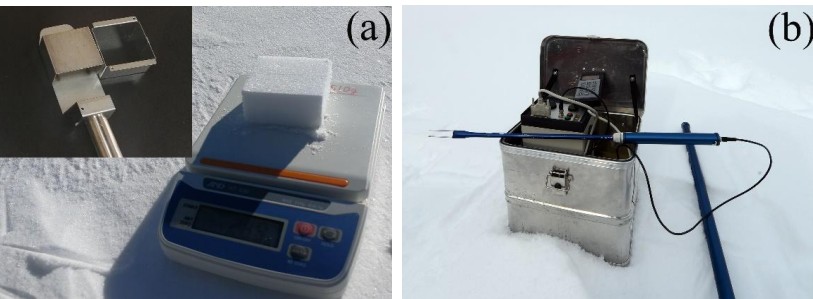


**Figure 1.** Snow density measurement systems: (a) the 100 cm³ box-cutter and the weighing scale,

(b) the Snow Fork measurement device.

Snow depth was the measurement of the vertical length of snow from ground to snow surface. The

ground was taken as reference datum recorded as 0 cm, and the vertical length from the ground to the

snow surface was recorded as the snow depth. Snow rulers were used to measure snow depth. We used

a brush to gently remove snow from the profiles, then examined the snow crystal diameter with the aid

of snow crystal screen which has two grids of 1 and 2mm under an 8 or 10×magnifier.

Shear strength is one of the fundamental mechanical properties of snow, which is the strength of

snow against the type of yield (Mellor, 1975; Nakamura et al., 2010). The shear strength of snow is

determined by microstructure and reflects the bond strength between the snow particles. The shear

frame system is usually used to measure the shear strength of snow (Mellor, 1975; Nakamura et al.,

2010). The measurement system consists of two parts: a shear frame with area of 0.025 m² and the

attached force gauges with full load capacity of 100 N (Fig. 2a) (Jamieson and Johnston, 2001, 2007;

Abe et al., 2006). Overlying snow was removed, leaving about 40mm of undisturbed snow above the

measured snow layer. The shear frame was inserted onto the measured snow layer and manually pulled

smoothly and quickly to ensure fracturing of measured snow layer. The shear strength is obtained by

dividing the force gauges data by effective shear area. In this study, the shear strength of the measured

layer was estimated by the shear strength from 12 shear measurements (Jamieson and Johnston, 2001,

2007; Abe et al., 2006; Nakamura et al., 2010).

Snow hardness indicates an ability of resistance to penetration or the pressure required for

penetration of snow (Pielmeier and Schneebeli, 2003). The higher the hardness, the smaller the

compressibility. Snow hardness was measured with the push-pull gauge with full load capacity of 100

N at observation sites (Fig. 2b). The attachment of the push-pull gauge was penetrated horizontally into

the snow profile at a uniform speed. When snow was completely covered with the attachment, the dial

readings were recorded. The snow hardness value $PR_{15}$ is the recorded data divided by the area of the

attachment, and the unit of $PR_{15}$ is Kpa. In this study, the hardness of the measured layer was estimated

by the hardness from 12 hardness measurements (Takeuchi et al., 1998).

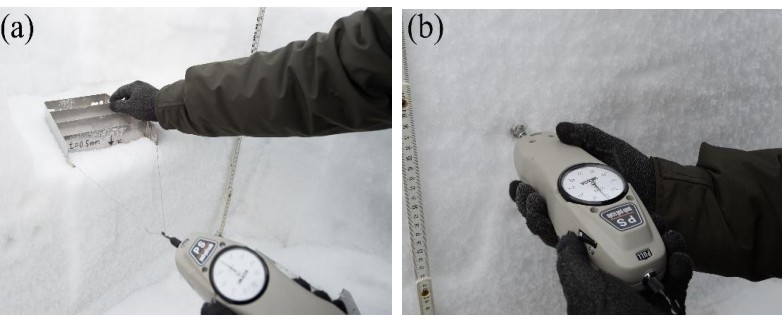

**Figure 2.** The measurements of snow mechanical properties: (a) snow shear strength measurement

(b) snow hardness measurement.

**2.3 Field experiment design**

To understand the performance of different measurement systems in different snow stratigraphies,

snow density data measured on different snow stratigraphical profiles by different measurement

systems were collected. Data for analysis were collected in the observation field of the TSSAR. Built in

1967, the TSSAR (43°16' N, 84°24' E) with an elevation of 1776m above sea level is located in the

upstream branch of the Künes River in the mid-mountain zone of the western Tianshan Mountains,

China. Based on observations at the TSSAR, the mean annual temperature is approximately 1.4°C, and

monthly minimum (January) and maximum (July) average temperatures are −17.7°C and 15.0°C,

respectively (Lu et al., 2016). Average annual precipitation is 870 mm and snowfall in the cold season

accounts for more than 30% of annual precipitation. Average snow depth in the station is 78cm with a

maximum snow depth of approximately 160cm in the cold season from 2000 to 2001. TSSAR has a

continental snow climate with snow water equivalent of less than 1000mm and average temperature of

less than -7℃ (Hao et al., 2018). The snowpack is characterized by lower snow density, lower snow

shear strength and high proportion depth hoar. High proportion depth hoar is considered to be the main

cause of frequent avalanches release in this area (Ma et al., 1990; Hao et al., 2018).

An experiment was designed to obtain profile density data measured at various snow

stratigraphies by different measurement systems. In order to obtain snow density data within various



stratigraphies, the experiment was carried out during different periods when the snow stratigraphy was significantly different. The experiment was first conducted on January 20th-21st, 2018, which represents a period of dry snowpack conditions. The depth and liquid water content of the snowpack were 55cm and 0～0.45% with the air temperature of -7.6℃ on the day, and the snowpack crystals consisted of decomposing and fragmented precipitation particles (DF) with 25%, faceted crystals (FC) particles with 33% and depth hoar (DH) particles with 42%. According to Fierz et al., (2009), and Techel and Pielmeier (2011), there were more high temperature gradient metamorphism grains (snow class FC, DH) than low temperature gradient metamorphism grains (snow class DF) in this snowpack. The experiment was performed again on March 10th, 2018, which represents a period of wet snowpack conditions. The depth and liquid water content of snowpack were 42cm and 3.0～6.0% with the air temperature of -0.7℃ on that day, and the snowpack crystals consisted of depth hoar (DH) with 8% and melt forms (MF) with 92%. The experimental method is described in the following section. Before implementing each density measurement experiment, a rectangular snow-pit was excavated in a previously undisturbed location and all measurements were made on a shaded side-wall of the snow-pit. The same experienced individuals took all samples and measurements.

**Measurements in dry and wet snowpack**: To compare the performance of different measurement systems in the same snowpack, whole dry snow layer density data by different measurement systems was collected on January 20th, 2018. The sampling was performed from the snow surface to ground at intervals of 30 mm and snow density was measured (Fig. 3a). The average density value of eighteen snow blocks from the snow surface to ground was calculated as the density of whole snow layer. The middle position of cutter snow sampling corresponded to the points of the Snow Fork measurement in order to compare the density from the two measurement systems at the same level (Fig.3). After one measurement of density within a whole snow layer, the observation position moved horizontally to next one. The density measurement of the whole snow layer was repeated with 20 times. Density data of whole wet snow layer by different measurement systems were collected on 10 March 2018. The same experimental method for measuring whole dry snow layer density was used to measurement the whole wet snow layer density. The average value of 14 snow blocks density from the snow surface to ground was calculated as the density of whole wet snow layer.

**Measurements within various snow stratigraphies**: to understand the performance of the same measurement system in different snow stratigraphies and compare the performance of different

measurement systems in various snow stratigraphies, the density data measured on the different dry

snow stratigraphical profiles by different measurement systems were collected on 21January 2018. The

shape and size of grain, and the depth of snow layers were first measured. After the stratigraphic

arrangement of the snowpack was identified (Table 1), the middle position of the given snow layer was

considered to be the target location of the measured density (Fig. 3b). Density measurements were

made in the DF, FC and DH snow layers by the two measurement systems and repeated with 25 times.

Shear strength and hardness ofeach layer were measured separately after the density measurements

were completed. Similarly, density, shear strength and hardness of the MF snow layer were measured

on 10 March 2018.

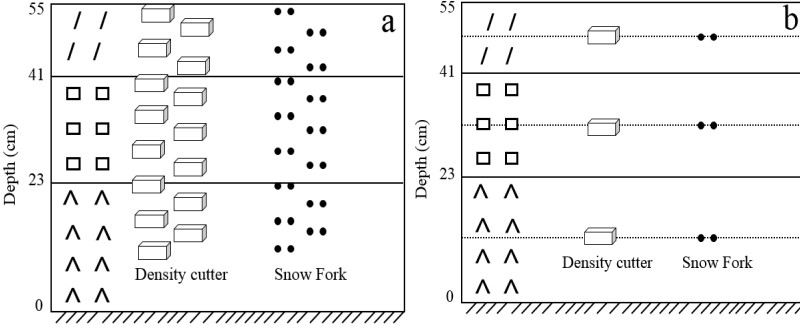

**Figure 3.** Schematic diagram showing the Snow Fork and the box-type cutter density measurements. (a)

Density measurement of the whole snow layer from the snow surface to ground. Intervals of the

measurements in the vertical direction are 30 mm. (b) Density measurement of snow layer made up of

the same type of grain. The slashes, hollow block and triangular brackets represent DF, FC and DH.

**Table 1.** Summary of sample layer characteristics and measurements.

| Snow layer | Grain shape | Layer thickness (cm) | Size of grain (mm) | Liquid water content (%) | Data | Number of samples |
|---|---|---|---|---|---|---|
| 1 | DF | 14 | 0.2~1 | 0~0.4% | 21 January 2018 | 25 |
| 2 | FC | 18 | 1~2 | 0 | 21 January 2018 | 25 |
| 3 | DH | 23 | 2~4 | 0 | 21 January 2018 | 25 |
| 4 | MF | 38 | 1~3 | 3.0%~4.5% | 10 March 2018 | 25 |

**2.4 Data analyses**

To understand and compare the performance of different measurement systems in various snow

stratigraphies, the accuracy and precision of the measurement system using density measurement data



were evaluated. The accuracy of a measurement system is the degree of closeness of measurements of a quantity to that quantity's true value (ISO, 1994). In this study, the average value of all samples in the same conditions was considered to the true value and was formulated by equation (1). Relative error (RE) and average relative error (ARE) were used to assess the accuracy of measurement system and

formulated by equation (2) and (3).

$$\bar{x} = \frac{1}{n} \sum_{i=1}^{n} x_i \qquad (1)$$

$$RE = \frac{x_i - \bar{x}}{\bar{x}} \times 100\% \qquad (2)$$

$$ARE = \frac{|RE|}{n} \qquad (3)$$

Where $\bar{x}$ denotes the average of all samples, $x_i$ is the density of the $i$ th sample, $n$ is the total

number of samples. The precision of a measurement system, related to reproducibility and repeatability, is the degree to which repeated measurements under unchanged conditions show the same results. Relative standard deviation (RSD) was used to assess the precision of each measurement system and formulated by equation (4). The smaller RSD, the higher the precision of a measurement system.

$$RSD = \frac{1}{\bar{x}} \sqrt{\frac{\sum_{i=1}^{n} (x_i - \bar{x})^2}{(n-1)}} \times 100\% \qquad (4)$$

Measurement uncertainty was a parameter characterizing the dispersion of the values attributed to a measured quantity. The higher the uncertainty, the lower the credibility and application value of data (BIPM et al., 2008). In this study, to evaluate measure uncertainty, the uncertainty ( $\mu_A$ ) was calculated according to the following formula (Elster, 2007; BIPM et al, 2008):

$$\mu_A = \sqrt{\frac{\sum_{i=1}^{n} (x_i - \bar{x})^2}{n(n-1)}} \qquad (5)$$

All statistical computations were implemented with statistical software (IBM SPSS Statistics 21). RE, ARE, RSD and $\mu_A$ were calculated for accuracy and precision of different measurement systems. The Levene's test was performed to verify departures from basic assumptions of variance and normality. A 1-way ANOVA was conducted to assess the overall statistical significance of differences among the measure groups (a=0.05).





**3 Results**

**3.1 Density measure of the whole snow layer**

The profiles of dry snow density obtained from two measurement systems are shown in Fig. 4(a).
The density of DH snow layer (0-23cm) was greater than that of DF and FC snow layers (23-55cm)
(Fig. 4(a)). For 23-55cm snow layers, the density of the ice layer (40-41cm) between DF and FC was
higher than that in other parts of the snow profile. The density of newly formed depth hoar layer
(21-23cm) was much lower than that of other parts of snow in the depth hoar layer. Significant
differences were found in the dry snowpack density of the box-cutter and the Snow Fork group
(P<0.01), and Table 2 showed that the average density of the whole dry snowpack was 187.3 kg m$^{-3}$
calculated from the box-cutter method and 169.4 kg m$^{-3}$ from the Snow Fork method. In other words,
whole dry snowpack average density of the box-cutter group was 1.1 times that of the Snow Fork group.
Statistically significant differences (P<0.05) were found in the ARE and the RSD of the two groups'
data (Table 2). The ARE and RSD of the box-cutter group were lower than that of the Snow Fork group.
Measurement uncertainty was 1.5 kg m$^{-3}$ calculated from the box-cutter method and 4.6 kg m$^{-3}$ from
the Snow Fork method in the dry snowpack.

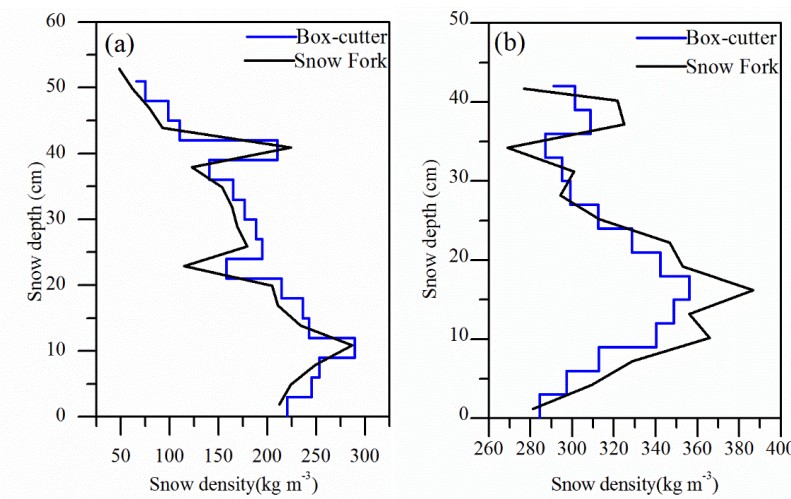


**Figure 4.** Density profile measured by different measurement system: (a) dry snow density data; (b)
wet snow density data.

The wet snow density profiles of the two measurement systems are shown in Fig. 4(b). The density
of snow from the 12cm to 18cm layer with liquid water content of 5%-6% was higher than that from
other layers. The density of snow from the box-cutter method was lower than that from the Snow Fork





method in the 12-18cm snow layer. The wet snow density of the box-cutter group was significantly

higher than that of the Snow Fork group (P<0.01), and the whole wet snowpack density of the Snow

Fork group was 1.03 times that of the box-cutter group. The ARE and RSD of the two groups' data in

wet snowpack were statistically significantly different (P<0.05) (Table 2). Measurement uncertainty

was calculated to be 1.9 kg m$^{-3}$ from the box-cutter and 3.3 kg m$^{-3}$ from the Snow Fork in the wet

snowpack. For both the dry and wet snowpack, the ARE, RSD, and measurement uncertainty of

measured data from the box-cutter group were lower than that from the Snow Fork group.

**Table 2.** Result summary of experimental measurements in field for the whole snow layer*

| Measurement system | Snow type | Layer mean density (kg/m$^3$) | RE (%) | ARE (%) | RSD (%) | $\mu_A$ (kg/m$^3$) |
|---|---|---|---|---|---|---|
| Box-cutter | Dry snow | 187.3 | -6～8 | 3.1 | 3.6 | 1.5 |
| | Wet snow | 311.5 | -4～5 | 2.3 | 2.7 | 1.9 |
| Snow Fork | Dry snow | 169.4 | -10～9 | 4.6 | 5.6 | 4.6 |
| | Wet snow | 320.2 | -8～9 | 3.6 | 5.1 | 3.3 |

* RE, relative error; ARE, average relative error; RSD, relative standard deviation; $\mu_A$, the

uncertainty.

### 3.2 Density measurement of the different snow layers

The density of different snow layers was measured by the box-cutter. The average density in DF,

FC, DH and MF layers measured by box-cutter was 93.5 kg m$^{-3}$, 177.3kg m$^{-3}$, 236.8 kg m$^{-3}$ and 304.9

kg m$^{-3}$ (Table 3), respectively. The results of measurement error and measurement uncertainty in Table

3 suggested that the ARE and RSD of different snow stratigraphies measured data were significantly

different (P<0.05) in the same box-cutter. For different snow layers, the ARE of measured data was as

follows: DF > FC > MF > DH. The ARE of DF layer measured were 4.2%, which was 2 times that of

DH layer measured. The box-cutter showed the highest accuracy in the DH, with a RE of -4%～3 %,

and an ARE of 2.0%. The order of RSD was as follows: DF > FC > MF > DH, and the RSD of DF

measured was 1.5 times that of DH measured. The accuracy and precision of the box-cutter were found

to be: DF <FC <MF <DH.

The average density of DF, FC, DH and MF layers measured by the Snow Fork was 68.7 kg m$^{-3}$,

168.3 kg m$^{-3}$, 224.3 kg m$^{-3}$ and 311.3 kg m$^{-3}$, respectively. The ARE and RSD of density measured by

the same Snow Fork were significantly different (P<0.05) in different snow layers. For different snow

layers, the ARE of measured was as follows: DF> FC > DH > MF (Table 3). The RE of the DF layer

measured was -18%～20%, which was about 2 times that of MF layer measured. The ARE of DH, MF



and FC layers measured was 4.1%, 2.9% and 3.4% respectively. The difference between the RSD measured from DH, MF and FC layers for the Snow Fork was significant (P <0.05). The RSD measured from DF layer was 18.9%, which was 4.9 times that of MF measured. The accuracy and

precision of the Snow Fork were found to be: DF <DH <FC < MF (Table 3).

Table 3 showed that the density in the DF, FC and DH layers from the box-cutter group were 1.36, 1.05 and 1.06 times that from the Snow Fork group, respectively. In contrast, the density in the MF layer from the Snow Fork group was higher than that from the box-cutter group, and the density from the Snow Fork group was 1.02 times that from the box-cutter group. The snow density from the

box-cutter and the Snow Fork showed the largest difference in DF layer, and the smallest difference in MF layer. The accuracy and precision of the box-cutter measurement system were higher than that of the Snow Fork measuring system in all four snow layers.

**Table 3** Summary of measurement system error, deviation and the uncertainty in different snow layers*.

| Measurement system | Grain shape | Layer mean density (kg/m3) | RE (%) | ARE (%) | RSD (%) | $\mu_A$ (kg/m3) |
|---|---|---|---|---|---|---|
| | DF | 93.5 | -8～10 | 4.2 | 4.9 | 0.9 |
| Box-cutter | FC | 177.3 | -4～6 | 2.9 | 3.4 | 1.2 |
| | DH | 236.8 | -4～3 | 2.1 | 3.2 | 1.6 |
| | MF | 304.9 | -6～7 | 2.8 | 3.3 | 2.0 |
| | DF | 68.7 | -18～20 | 14.7 | 18.9 | 2.6 |
| Snow Fork | FC | 168.3 | -9～11 | 4.1 | 5.3 | 1.7 |
| | DH | 224.3 | -10～9 | 3.4 | 4.2 | 1.9 |
| | MF | 311.3 | -7～9 | 2.9 | 3.8 | 2.4 |

*RE, relative error; ARE, average relative error; RSD, relative standard deviation; $\mu_A$, the uncertainty.

**4 Discussion**

**4.1 Performance of gravimetric measurement systems in various snow stratigraphies**

Natural snowpack develops from a series of winter snowfall and contains many layers with different characteristics (Sturm et al., 2002; Kärkäs et al., 2005). The snow stratigraphic layers were

defined by size of the grains and microstructure (Fierz et al. 2009). This study analyzed experiments where the same measurement system was used to measure the density of the different snow stratigraphies to assess whether the same measurement system showed similar performance in different snow stratigraphies. Field experimental reports showed that the same measurement system had significantly different accuracy and precision in various snow stratigraphies (Table 2). For different

snow stratigraphies in dry snow, the accuracy and precision of the box-cutter showed as follows: DF

(93.5 kg m$^3$) < FC (177.3 kg m$^3$) < DH (236.8 kg m$^3$). The box-cutter had poor accuracy in lower

density snow layers, which was in agreement with findings by Conger and McClung (2009), who found

that the accuracy of 100cm$^3$ box-cutter in high density snowpack was higher than that in low density

snowpack. Carrol (1977) and Proksch et al. (2016) also reported that the box-cutter had poor accuracy

and tends to overestimate density for lower density snow.

Snow has characteristics of easy compression and deformation (Sturm and Holmgren, 1998). Snow

samples are compressed and sheared by the cutter and cover in the process of sampling, which can

cause the densification of snow samples. This may result in density being overestimated. The degree of

compression deformation of the snow sample is affected by the force and velocity of the thrust of the

box into the snow. The force and speed of propulsion cannot be constant in each measurement in the

field. Therefore, snow with greater compressibility is more susceptible to artificial manipulation,

resulting in the measurement system having high system error and measurement uncertainty. A similar

conclusion was reached by Hawley et al. (2008) who reported that unconsolidated and low density

snow near the snow surface affects the accuracy measurement of the measurement system, but Hawley

et al. (2008) did not further discuss and explain the reasons in detail.

It is difficult to quantify the compressibility of snow in the field, so we use hardness as an indicator

(Pielmeier and Schneebeli, 2003). Snow compressibility decreases with the increase of the hardness.

Experimental observations found hardness and the shear strength of snow layer in the following order:

DH > FC>DF in dry snow (Fig. 5). This result can be caused by the fact that compressibility decreases

with the increase of the size of grain and density in dry snow (Sturm and Holmgren, 1998), and the

shear strength of snow is higher when snow density is greater in dry snow (Abe and other, 2006). DF

was located at the top of the whole snow layer (Fig. 3) and DF had low temperature gradient

metamorphism grains (Fierz et al., 2009; Techel and Pielmeier, 2011) with low density and small grain

size (Table 1). Low density and small grain size of DF caused low shear strength and high

compressibility (Fig. 5a), resulting in high susceptibility to compressive deformation and shear failure

from external force. The degree of compressive deformation and shear failure varied greatly with each

measurement for DF, so that measured value showed high disparity and dispersion. Therefore, the

poorest accuracy and precision of the measurement system was found in DF. DH was located at the

bottom of the whole snow layer, and its density and size of grain were the largest (Table 1). High





density and large gain size of DH caused high shear strength and low compressibility, resulting in

measured values with relatively low disparity and dispersion. Across the four dry snow layers, the

measurement system showed the highest accuracy and precision in DH due to high shear strength

low compressibility which is a characteristic of DH. From the short review above, the accuracy and

precision of measurement system showed increased tendencies with increasing density and size of

grain of snow, and the accuracy and precision of measurement system showed as follows: DF < FC <

DH in dry snow.

In contrast, the density of MF was higher than that of DH, but the accuracy and precision of

measurement system in MF were lower than that of DH (Table 3). MF has isothermal gradient

metamorphism grains (Fierz et al. 2009), and the liquid water content of MF layer was 3.0%～4.5%

(Table 3). The shear strength of snow showed decreasing tendencies with the increase of liquid water

content (Yamanoi and Endo, 2002). Although the density of MF was higher than that of DH,

experimental observation found the shear strength and hardness of MF were lower than that of DH (Fig.

5). This result can be explained by the fact that the accuracy and precision of the measurement system

in MF were lower than that of DH. In summary, the native density, liquid water content, and the size of

grain of four different snow layers caused the decrease of hardness and shear strength of the snow

layers in the following order: DH >MF>FC>DF (Fig. 5), which led to the reduced accuracy and

precision of the measurement system in the same order. However, the study also found that the

measurement system showed relatively low accuracy and precision in the ice layers and the bottom

depth hoar layer which have relatively high density. Because ice layers and the bottom depth hoar have

high shear strength and hardness, box-cutter and cover had difficulty cutting snow and a part of the

snow sample was lost in the course of sampling.



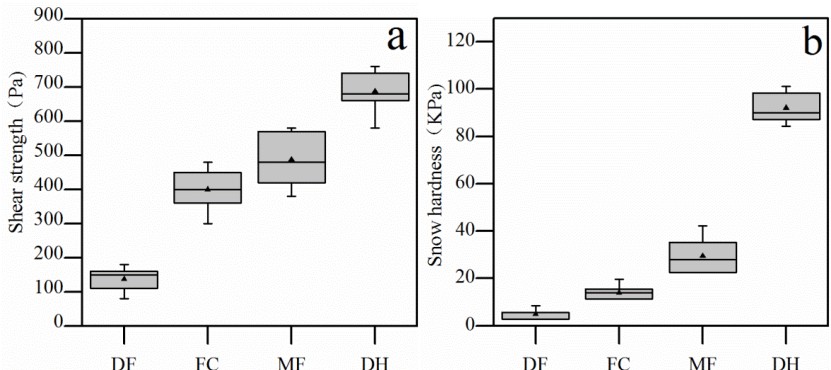

**Figure 5.** The mechanical properties of the target snow layer: (a) the shear strength of the target snow layer; (b) the hardness of the target snow layer. Boxes represent interquartile ranges (25th to 75th percentiles), thick horizontal bars in each box denotes the median (50th percentile), Whiskers (vertical lines and thin horizontal bars) represent the highest and lowest value within 1.5 times the interquartile range above the upper or below the lower quartile, respectively. Black small triangle denotes the average values.

**4.2 Performance of dielectric permittivity measurement systems in various snow stratigraphies**

Dielectric permittivity measurement systems are based on the principle of measuring the response characteristics (travel time, reflection, attenuation) of an electromagnetic signal through snow (Frolov and Macheret, 1999). The ratio of water, air, and ice in snow determines the dielectric permittivity of snow (Tiuri and Sihvola, 1984; Schneebeli et al., 1998). The snow density is calculated using the electrical properties of ice, water, and air measured by the dielectric permittivity measuring instrument (Tiuri et al., 1984; Kovacs et al., 1995). The accuracy and precision of the measurement system in low density snow (snow class DF) were lower than that in high density snow (snow class FC, DH and MF) (Table 3). In terms of the accuracy of dielectric permittivity measurement system, a similar result was showed by Hawley et al. (2008) who found RE up to 20% in the low-density sections and 10–13% in the high density sections for the permittivity measurement system. Therefore, dielectric permittivity measurement system was extremely unstable in low density snow.

There are several other interesting facts worth discussing based on the findings of the field experiment. Hawley et al. (2008) thought that low density snow had a bigger air gap between the grains resulting in unstable capacitance readings. Sugiyama et al. (2010) reported that the structure of snow affected the stability of dielectric signals and the permittivity of unconsolidated snow was smaller than





that of compacted snow. However, they did not consider how the damage of the resonator to the snow structure would affect the accuracy and precision of dielectric permittivity measurement system. Snow sampling was not compressed in dielectric permittivity measurement, but snow structure was damaged when the resonator was inserted horizontally into the snow profile. The destruction resistance

capability of snow affected the accuracy and precision of the measurement system. Disparity and dispersion of the measured value increased with the increasing degree of damage to the snow structure from the resonator. Because low density snow (snow class DF) had relatively low destruction resistance capability due to low shear strength and hardness (Fig. 5), the measurement system showed poor accuracy and precision in low density snow. On the contrary, although the destruction resistant

capability of MF was lower than that of DH, the accuracy and precision of the measurement system in MF were slightly higher than that of DH. This can be explained by liquid water filling in the gap between the grains in the MF layer, resulting in relatively small air gap between the grains, which further led to relatively stable dielectric signals of the measurement system in the MF layer. Temperature of the snow layer and sand-dust in snowpack also affect the dielectric permittivity of snow

(Fujita et al., 1992; Dong et al., 2014). There were significant differences in temperature and sand-dust between snow layers. Because of the lack of conditions for measuring temperature and sand-dust in the field, we did not take the effect of temperature and sand-dust on the measurement accuracy and precision into consideration. Understanding the effects of these factors on density measurement would be worthwhile in future research projects.

**4.3 Comparison of two measurement systems**

There was significant difference in the snow density observed from the two measurement systems, and the difference varied with changes in snowpack characteristics. For dry snow, because snow was compressed in the process of cutter sampling, the density data from the box-cutter were much greater than that from the Snow Fork (Table 2). The whole dry snowpack density from the box-cutter was

higher than that from the Snow Fork by 10%, expressed as percentage of the mean Snow Fork density (Table 2). Leppänen et al. (2016) also reported that the snow density from the Snow Fork was lower than that from the gravimetric measurement system. The density of DF, FC and DH from the box-cutter overestimated that from the Snow Fork by 34%, 5%, and 6 %, respectively (Table 3). The density differences between the two measurement systems decreased with the increase in density. The same

result was reported by Hawley et al. (2008) who found the dielectric permittivity measurement system





underestimated density in the lower-density sections but was close to the gravimetric measurement system for the higher-density sections in dry snow. For the whole wet snowpack, the density from box-cutter was slightly lower than that from the Snow Fork (Table 2). The density from the box-cutter measurement system underestimated that from the Snow Fork measurement system by 3% in the MF

snow layer (Table 2). In summary, the box-cutter appeared to overestimate snow density in the lower-density sections with respect to the Snow Fork densities but agreed with or slightly underestimated the higher-density sections. A similar phenomenon was presented in the comparison of micro-computed tomography (μCT) and box-cutter, where the box-cutter tended to overestimate low density sections and underestimate high density sections with respect to the μCT densities in the field

(Proksch et al., 2016).

The accuracy and precision of the box-cutter and the Snow Fork showed lower accuracy and precision in low density snow (snow class DF) due to lower shear strength and hardness of low density snow. The difference in the accuracy and precision of the two measurement systems showed decreasing tendencies with increasing density. At present, from the perspective of accuracy and precision only, the

box-cutter performed better than the Snow Fork in the field, especially in low-density snowpack, where the Snow Fork showed very poor accuracy and precision. The weight of the box-cutter (box-cutter, flat shovel and balance scales) was lighter than the Snow Fork (Table 4). The box-cutter was also easier for surveyors to carry in the cold environment. Although some of snow escaped from the sampler or stays in the sampler during the transfer from sampler to weighing scales causing some errors, these error

sources are minimized by carefully cleaning the sampler and weighing scales during the measurements. Therefore, based on the above analysis, the box-cutter was more suitable for field snow density measurement than the Snow Fork. However, the vertical resolution of the Snow Fork in the millimeter range was clearly a significant advantage on the centimeter resolution of the box-cutter (Table 4). The box-cutter was usually insufficient to resolve the small-scale spatial fluctuations in density so it may

not meet the needs of some research. Measurement time of the Snow Fork was less than the box-cutter in the cold environment. The surveyor needed an assistant to record data in the box-cutter system measuring, but the Snow Fork could automatically save data (Table 4). The most remarkable advantage of the Snow Fork over the box-cutter was that the Snow Fork system can measure density and obtain liquid water content in snowpack at the same time in the wet snowpack.






**Table 4**. Vertical resolution and weight of the different measurement system. Measurement time in the field is per meter of snow depth and includes digging a snow pit.

| Measurement system | Vertical resolution (mm) | Weight (kg) | Measurement time | Cost (US dollar) | Data record |
|---|---|---|---|---|---|
| Box-cutter | 30 | 21.0 | 60min | 20 | Manual records |
| Snow Fork | 3 | 1.3 | 40min | 14500 | Automatic records |

Snow density from the two measurement systems was significantly different. The box-cutter either over- or underestimated the snow density relative to the Snow Fork under different snow conditions.

Different measurement systems might be used in field snow measurement due to different study conditions, such as labour needs, data requirements, and different research teams and institutions. Therefore, it was difficult to build a unified and optimized snow global database. Assimilation of the data obtained by different measurement systems provided an effective way to solve the above problem. The correlation of the density from box-cutter system ($\rho_b$) with the density from the Snow Fork

system ($\rho_s$) at the same snowpack was investigated in the dry and wet snow (Fig. 6). Least-squares linear regression gave the coefficients of the relationship

$$\rho_b = a\rho_s^2 + b\rho_s \qquad (6)$$

as $a = -0.11 \times 10^{-2}$ and $b = 130.75 \times 10^{-2}$ with $R^2 = 0.95$ in the dry snow (Fig. 6a), and as $a = -0.07 \times 10^{-2}$ and $b = 123.8 \times 10^{-2}$ with $R^2 = 0.88$ in the wet snow (Fig. 6b).

The density from box-cutter was greater than that from the Snow Fork in the dry snowpack. Whereas the density from box-cutter tended to overestimate low densities and underestimate high densities with respect to the density from the Snow Fork, with a threshold of 318 kg m$^{-3}$ in the wet snowpack (Fig. 6b).

The correlation of the density from box-cutter and the Snow Fork at the same snowpack was

investigated using all the data obtained from this study (Fig. 7). Least-squares linear regression gives the coefficients of the relationship as $a = -0.08 \times 10^{-2}$ and $b = 126.13 \times 10^{-2}$ with $R^2 = 0.97$.

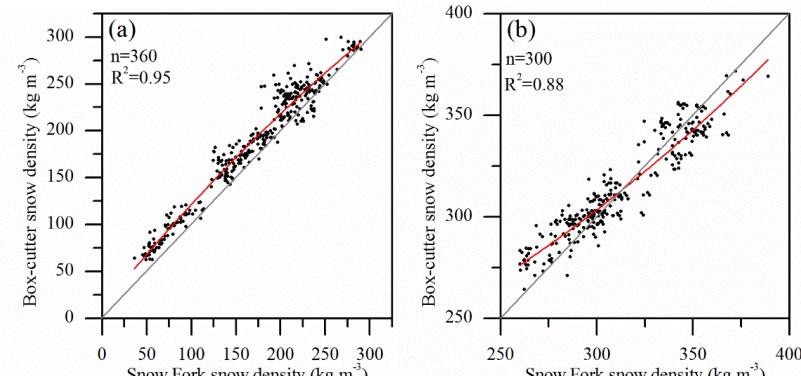

**Figure 6.** Box-Cutter density vs Snow Fork density. The lines correspond to the linear regression of the two datasets (a) dry snow; (b) wet snow.

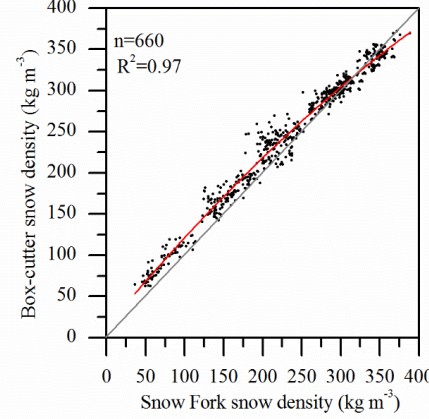

**Figure 7.** Plots of all the data on the permittivity vs snow density. The lines correspond to the linear regression of the two datasets.

**5 Conclusions**

This study compared snow density measured by the box-cutter and the Snow Fork measurement systems in the TSSAR in the winter of 2017-2018. The results showed that the snow density from two measurement systems was significantly different in the same snowpack. The snow density from the box-cutter system tended to overestimate low density and underestimate high density with respect to the snow density from the Snow Fork system. The density in layers consisting of low temperature gradient metamorphism grains (snow class DF) was more frequently falsely estimated than that in layers consisting of high temperature gradient metamorphism grains (snow class FC, DH) or isothermal gradient metamorphism grains (snow class MF) in all measurement systems.

The accuracy and precision of density measurement systems are critical for experiments and





observations of snow. Each system of density measurement has its own advantages and limitations. The two measurement systems showed poor accuracy and precision in low-density dry snow. Both measurement systems need further refinement for use on low density snow (snow class DF). The accuracy and precision of the box-cutter were better than that of the Snow Fork in field observation. Although the box-cutter is generally a more reliable method for density measurement, its resolution is

low and its damage to snow profile is more serious than Snow Fork. Furthermore, there was a slight difference between the Snow Fork densities and box-cutter densities in wet snowpack, and the Snow Fork can obtain liquid water content data while measuring density. Considering the accuracy, cost, labour needs, and data requirements, the study results suggest snow surveyors use box-cutter systems during dry snow and the Snow Fork systems during wet snow in the investigation of snow

characteristics. This will help surveyors obtain more reliable data and optimize field measurements.

Our study could provide an approach which will aid researchers in assimilating data from separate studies obtained with different measurement systems and integrating single studies into larger databases. However, the lack of data from other measurement systems (μCT, neutron-scattering probe, other type cutters, etc) makes it difficult to compare and evaluate the performance of other

measurement systems across the different snow stratigraphies. The data collection from other measurement systems can help to build a better data assimilation scheme in the future.

*Acknowledgements.* This work was supported by the National Project of Investigation of Basic Resources for Science and Technology (Grant No. 2017FY100501), the National Natural Science

Foundation of China (NSFC Grant U1703241) and CAS Instrumental development project of Automatic Meteorological Observation System with Multifunctional Modularization. We are grateful to the supports in field and laboratory work from the Tianshan Station for Snowcover and Avalanche Research, Chinese Academy of Sciences.

*Author Contributions:* Jiansheng Hao conceived and designed the study, and wrote the paper. Jiansheng

Hao, Farong Huang, Ditao Chen performed the experiments. Lanhai Li and Shuyong Mu contributed to discussions and revisions.

*Conflicts of Interest:* The authors declare that they have no conflict of interest.

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
