# Peer review of "Performance of snow density measurement systems in"

_Geoscientific Instrumentation, Methods and Data Systems, 2020_

## Short Comment (SC1) · 18 Jun 2020

During the comparison of two techniques in the analysis of material or substance, how we tell that which one is better?.So first, we should choose standard reference material,or location and analyse first this one. Then by knowing the performance of which technique is better at standard and we should go to experiment, with two techniques. finally we have to conclude the performance of one technique out of two through comparison results of particular location or sample with previous standard results.

---

## Author Comment (AC1) · 18 Jun 2020

Data availability statement: The data related to this article is available online at https://doi.org/10.5194/gi-2020-14-supplement.

Please also note the supplement to this comment:
https://gi.copernicus.org/preprints/gi-2020-14/gi-2020-14-AC1-supplement.zip

---

## Referee Comment (RC1) · Xuemei Li (Referee) · 26 Jun 2020

Snow density is a fundamental property of snowpack and plays a key role for a wide range of applications and almost all of them require density values. Snow hydrology and climatology require snow density. A precise measurement of snow density and its variation in horizontal and vertical directions is of major importance to better understand and model a wide range of snow physical processes. Snow density varies with crystal size, shape, and the degree of riming during a snow season. Snow density is a complex parameter that can vary spatially, temporally and vertically within the snow pack profile. A precise measurement of snow density has been sought for a long time, and various measurement systems are used to obtain snow density so as to produce many data sets with significant differences. The idea of this research is novel,

especially the influence of snow hardness on different measuring systems. The result will help us to choose more effective and reasonable measurement system based on snowpack characteristics in the field, and will give us a better understanding of the snow characteristics of the instrument for carrying out snow hydrology research. The study has important scientific significance and practical value. However, there are few details that need to be revised before accepting this article. 1.The description of the global snow density database and spatial and temporal variability in seasonal snow density refers to Bormann (2013). Bormann, K. J., Westra, S., Evans, J. P., and Mccabe, M. F.: Spatial and temporal variability in seasonal snow density, J. Hydrol., 484, 63-73, https://doi.org/10.1016/j.jhydrol.2013.01.032, 2013. 2.The influence factors on the accuracy of density measurement system are described and studied by referring to López‐Moreno (2020) and Kaur (2017). Kaur, S., and Satyawali, P. K.: Estimation of snow density from SnowMicroPen measurements, Cold Reg. Sci. Technol., 134, 1-10, https://doi.org/10.1016/j.coldregions.2016.11.001,2017. López‐Moreno, J.I., Leppänen, L., Luks, B., Holko, L., Picard, G., Sanmiguel‐Vallelado, A., Alonso‐González, E., Finger, D.C., Arslan, A.N., Gillemot, K. and Sensoy, A.: Intercomparison of measurements of bulk snow density and water equivalent of snow cover with snow core samplers: instrumental bias and variability induced by observers, Hydrol. Process., https://doi.org/10.1002/hyp.13785, 2020. 3.In the section of "2.1 Measurement systems of snow density", The permittivity is measured directly by the snow fork instead of the snow density. The author needs to supplement the principle and relevant formula of obtaining snow density based on permittivity. 4. P7: Too much information about the environment of the experimental observation site is introduced. Much of the information is irrelevant to this study. 5.The performance of different measuring systems under dry and wet snow conditions was compared. But a detailed definition of dry and wet snow was not given in the study. This makes it difficult for field observers to refer to the results. 6.A discussion of the causes of the error of the gravimetric measuring systems refer to a recently published paper López‐Moreno (2020). 7.Some editing and proofreading need.

---

## Referee Comment (RC2) · Anonymous Referee #2 · 6 Jul 2020

The authors carried out the comparison of snow density observations between gravimetric and dielectric permittivity measuring systems under dry and wet snow conditions. Then, they analyzed these data with considering characteristic of snow layer stratigraphies. Based on these analyses results, they discussed the causes of dependency of measurement error on its snow layer stratigraphy. Finally, they recommend which measurement system is better for snow density measurements.

First of all, it is clear to see that a lot of hard work has been put into the study. Measurement data have enough number statistically, therefore, I do not doubt their statistical analyses results.

However, the overall investigation is not novel from the scientific viewpoints and conclusion addressed by this paper is a little bit weak under the present version for publication

of G.I.

For improvement of the manuscript, I point out the following major comments and specific comments:

<Major comments>

1. The logic to compare different two methods is ambiguous. When we discuss the relative accuracy between different methods, we firstly need to determine the standard reference method, then compare the measured data of different methods with measured data of the standard reference method, but the authors do not do it. For clearing the logic, the standard reference method should be defined in the manuscript.

2. The organization of the manuscript leaves plenty of room for improvement because similar descriptions appear at several paragraphs and some unimportant information were included. For the level of publication, reconsider the organization to show the scientific significance of the work efficiently.

<Specific comments>

<2.3. Field experiment design>

L206: Were the values of water contents (0-0.45%) true? Air temperature was so low (-7.6 degree Celsius) that snow could not be wet. Including this part, the authors should consider significant figures obtained from the equipment through the manuscript.

Table 1: In the Fig. 3, there is not a MF layer. If the authors obtained the MF layer form the same snow pits shown in Fig. 3, please show it in the figure.

<Results>

L296-297: Wet snow density of the box-cutter group should be lower than that of the Snow Fork.

<Discussion>

L371: Although the authors insisted that "the degree of compressive deformation and shear failure varied greatly with each measurement for DF", Fig. 5 does not show such trend. Please indicate which figure shows this result.

L406-410: The explanation of Snow Fork is not suitable in Discussion part. They should be moved to the Method part.

L431-433: I can not understand their logic. Please add more detail explanation to support their arguments.

L442-444: L452-455: Their arguments are not convincing because there is no evidence that the densities of Snow Fork measurement are correct. As mentioned in Major comment, the authors must clear which measurement method is standard and then compare each other if they want to discuss the superiority or inferiority.

L466-467: In the Table 4, weight of box-cutter is much heavier than Snow Fork although they claimed that weight of box-cutter is lighter than Snow Fork.

---

## Author Comment (AC2) · 3 Aug 2020

Thank you very much for your timely feedback of our manuscript. We are very grateful to your critical comments and thoughtful suggestions. Those comments and suggestions are all valuable and very helpful for revising and improving our paper, as well as guiding our further research. Based on these comments and suggestions, we have made careful modifications in the original manuscript. Please find our responses to reviewer and proposed changes in the manuscript in the attached file.

Please also note the supplement to this comment:
https://gi.copernicus.org/preprints/gi-2020-14/gi-2020-14-AC2-supplement.zip

---

## Author Comment (AC4) · 3 Aug 2020

We are very grateful to your critical comments and thoughtful suggestions. Those comments and suggestions are all valuable and very helpful for revising and improving our paper. We apologize for the confusion generated by original version of the manuscript and sincerely hope that our logic is now easier to follow with this revised version. We revised the logic, and reorganized and rewrote both results and discussion sections. The logic of new version: The average density from gravimetric permittivity measuring systems was defined as the reference truth value in this study. Then the data characteristics of various snow stratigraphies from gravimetric and dielectric permittivity measuring systems were analyzed. Performance of gravimetric and dielectric permittivity measuring systems and error analysis in various snow

stratigraphies were discussed. Take gravimetric measuring method as the standard reference method, the precision and accuracy of dielectric permittivity measuring systems were compared. We completely rewrote both results based on the new logic and reorganized discussion sections. You will find new version in supplement.

Please also note the supplement to this comment:
https://gi.copernicus.org/preprints/gi-2020-14/gi-2020-14-AC4-supplement.zip